# HermesFlow: Seamlessly Closing the Gap in Multimodal Understanding and Generation

**Ling Yang**[1]*† **Xinchen Zhang**[2]* **Ye Tian**[3] **Shiyi Zhang**[2] **Chenming Shang**[2]
**Minghao Xu**[3] **Wentao Zhang**[3]† **Bin Cui**[3]†
[1] Princeton University  [2] Tsinghua University  [3] Peking University
https://github.com/Gen-Verse/HermesFlow

## Abstract

The remarkable success of the autoregressive paradigm has made significant advancement in Multimodal Large Language Models (MLLMs), with powerful models like Show-o, Transfusion and Emu3 achieving notable progress in unified image understanding and generation. For the first time, we uncover a common phenomenon: the understanding capabilities of MLLMs are typically stronger than their generative capabilities, with a significant gap between the two. Building on this insight, we propose **HermesFlow**, a simple yet general framework designed to seamlessly bridge the gap between understanding and generation in MLLMs. Specifically, we take the homologous data as input to curate homologous preference data of both understanding and generation. Through Pair-DPO and self-play iterative optimization, HermesFlow effectively aligns multimodal understanding and generation using homologous preference data. Extensive experiments demonstrate the significant superiority of our approach over prior methods, particularly in narrowing the gap between multimodal understanding and generation. These findings highlight the potential of HermesFlow as a general alignment framework for next-generation multimodal foundation models.

## 1 Introduction

The rapid advancement of Large Language Models (LLMs) [24, 9, 55, 53, 74] has driven significant development in both multimodal understanding [22, 72, 19] and autoregressive image generation [34, 38, 4]. Recent studies [36, 18, 45, 44, 23] focused on developing a unified system capable of both multimodal understanding and generation. Powerful Multimodal Large Language Models (MLLMs) like Show-o [48], Transfusion [69], and Emu3 [41], employ a single transformer to unify these tasks, demonstrating remarkable performance across both domains.

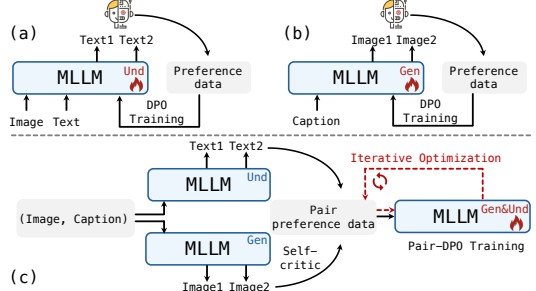

Figure 1: Architecture comparison between (a) DPO training improve multimodal understanding [71, 11], (b) DPO training improve multimodal generation [41] and (c) our HermesFlow.

Recently, there has been growing interest in exploring the synergy between multimodal understanding and generation [45, 40, 3]. Liquid [45] demonstrates that these two tasks are mutually

---

*Contributed equally.
†Corresponding authors.

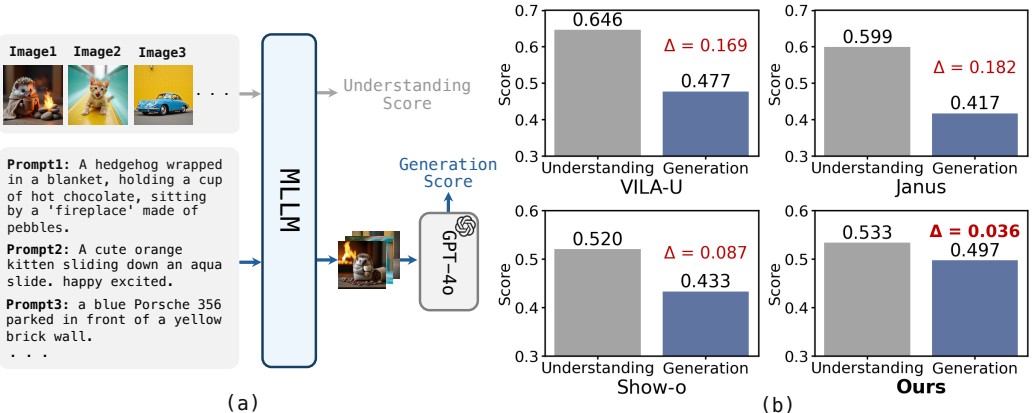

Figure 2: **Motivation of HermesFlow.** (a) A general pipeline to quantitatively assess the MLLM's performance of multimodal understanding and generation. (b) The imbalance between understanding and generation capabilities is a common phenomenon in MLLMs, and our method significantly narrows this disparity. For detailed descriptions, please refer to Section 5.2.

beneficial: expanding the data for either understanding or generation enhances the performance of the other. Furthermore, MetaMorph [40] reveals that understanding data is more effective than generation data in improving both understanding and generation performance. However, these works jointly improve the understanding and generation capabilities of MLLMs from a data-level perspective but fail to consider the gap between them. It remains unclear whether a capability gap exists between them.

Regarding both task difficulty and architectural constraints, there exists a significant gap between multimodal understanding and generation. In terms of task difficulty, generation maps textual features to visual domain, requiring both semantic accuracy and high-fidelity details. However, converting compressed text into high-dimensional visuals inherently loses features, making generation harder. In contrast, understanding tasks involve compressing rich visual information into textual domain, typically focusing on low-frequency semantics. It is important to clarify that visual understanding is a broad concept; in this paper, we specifically refer to the basic feature understanding of natural images. Accordingly, generation is fundamentally more challenging. For architectural constraints, image generation struggles with autoregressive modeling because visual spatiality conflicts with sequential token prediction, yielding suboptimal results. In contrast, understanding tasks benefit from the natural alignment with autoregressive text serialization. Thus, generation also encounters greater architectural challenges.

To quantitatively assess the performance of multimodal understanding and generation, we design a general pipeline, as illustrated in Figure 2 (a). For any pretrained MLLM, input consists of (image, prompt/caption) pairs. For understanding tasks, MLLM is presented with multiple questions related to each image, and the final understanding score is calculated as the average accuracy of its answers. MLLM generates an image for each prompt, and these images are evaluated by posing the same set of questions using GPT-4o [16], with the final generation score calculated based on the average accuracy of GPT-4o's answers. We employed this pipeline to evaluate multiple MLLMs. As demonstrated in Figure 2 (b), unified models like VILA-U [47], Janus [44] and Show-o [48] exhibit notably stronger understanding capabilities compared to their generation capabilities. Our experiments highlight a recurring phenomenon: **MLLMs consistently demonstrate superior understanding abilities over generation abilities, with a significant gap between them**.

In the pretraining of MLLMs, simply increasing the training data for understanding or generation does not yield proportional improvements in both aspects [40], leaving a significant gap between their understanding and generation capabilities. A truly unified MLLM should excel in both understanding and generation, striking a balance rather than favoring one over the other. However, current unified MLLMs struggle to achieve this equilibrium. To bridge the gap between understanding and generation in MLLMs, we propose *HermesFlow*, a self-improvement framework that collects paired understanding and generation preferences from homologous input data, and then employ a novel Pair-DPO post-training framework to seamlessly bridge the gap through the paired preference data. To curate understanding preference data, we enable MLLM to generate multiple captions for a single

input image and filter paired understanding preference data using BERT similarity scores. To curate generation preference data, we prompt MLLM to generate multiple images from a single prompt and employ a self-critic-like approach to evaluate the images through self-VQA scoring, thereby filtering and selecting the paired generation preference data. Finally, we design Pair-DPO for preference alignment of homologous paired data, and through iterative optimization to simultaneously and progressively reduce the gap between understanding and generation following the same approach. We achieve the self-improvement of both understanding and generation of MLLM without incorporating any external high-quality training data. As shown in Figure 2 (b), based on Show-o, HermesFlow not only reduces the gap between understanding and generation, but also improves both capabilities, demonstrating the necessity of bridging the gap os these two abilities.

We compare HermesFlow with previous work in Figure 1 and summarize our contributions as follows:

- An insightful discovery regarding a significant gap between the understanding and generation abilities of MLLMs, with understanding consistently outperforming generation.

- We propose a general multimodal self-improvement framework, *HermesFlow*, using Pair-DPO based on homologous data to seamlessly close the gap between multimodal understanding and generation.

- Self-play iterative optimization paradigm is highly compatible with the multi-round enhancement of MLLMs. HermesFlow has potential as a general alignment framework for next-generation multimodal foundation models.

- Extensive qualitative and quantitative comparisons with previous powerful methods, such as Show-o, Janus and VILA-U, demonstrate the effectiveness and superiority of our method.

## 2 Related Work

### 2.1 Unified Multimodal Understanding and Generation

In recent years, a growing number of studies [3, 6, 46, 57, 23, 32] have explored unified multimodal models capable of both visual understanding [10, 62, 66, 31, 51] and generation [65, 52]. Early methods [3, 40, 6, 35, 73, 63] leveraged diffusion models as external tools, where MLLMs generate conditions for visual generation [54, 39] without having direct generative capabilities. For instance, DreamLLM [3] introduces learnable embeddings called dream queries, which encapsulate the semantics encoded by MLLMs and serve as conditions for the diffusion decoder. More recently, inspired by the success of autoregressive paradigms, many studies [36, 48, 69, 26, 49, 59, 1, 61, 41] have shifted focus to representing and generating images using discrete visual tokens within a single transformer framework. For instance, Emu3 [41] is trained solely with next-token prediction on a mixture of multimodal sequences using a single transformer. Janus [44] separates visual encoding into distinct pathways for multimodal understanding and generation while maintaining a unified transformer architecture. However, no existing research has focused on the relationship between the strengths of understanding and generation capabilities in MLLMs, which is essential for the balanced and sustainable development of these models.

### 2.2 DPO in Multimodal LLMs

Direct Preference Optimization (DPO) [28, 64, 56, 53] enhances the performance of multimodal LLMs through the post-training process. In Figure 1, we categorize these approaches into three types. Some methods [70, 71, 11, 60] utilize DPO to enhance understanding capability, as shown in Figure 1 (a). For instance, CSR [71] enables the model to self-improve by iteratively generating candidate responses, evaluating the reward for each response, and curating preference data for finetuning. Other methods [41] improve the generation capability of MLLMs through DPO as illustrated in Figure 1 (b). Emu3 [41] generates a data pool and constructs a preference dataset through manual ranking, which is then used to optimize the model's generation capabilities via DPO. However, these models focus exclusively on enhancing either understanding or generation capabilities. In contrast, our approach uses Pair-DPO to effectively narrow the gap between the two, achieving mutual improvement.

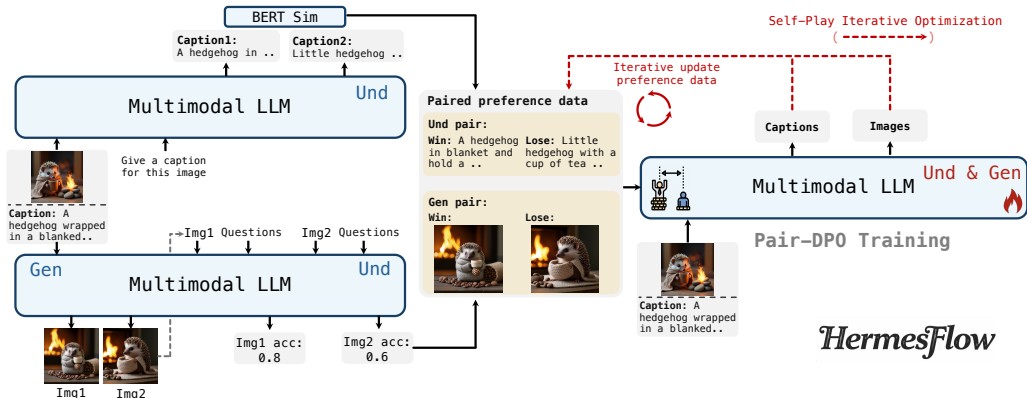

Figure 3: **Pipeline of HermesFlow.** We begin by curating paired data that captures both understanding and generation preferences from homologous input data. Leveraging this homologous preference data, we design Pair-DPO and employ self-play iterative optimization to seamlessly bridge the gap between multimodal understanding and generation.

## 3 Preliminary

### 3.1 Next Token Prediction

Next token prediction is a fundamental task in sequence modeling, where the goal is to estimate the conditional probability of the next token $x_t$ given its preceding context $x_{<t} = \{x_1, x_2, \ldots, x_{t-1}\}$. Formally, for a sequence $\mathbf{x} = \{x_1, x_2, \ldots, x_T\}$, the joint probability is factorized as:

$$P(\mathbf{x}) = \prod_{t=1}^{T} P(x_t | x_1, x_2, \ldots, x_{t-1}) = \prod_{t=1}^{T} P(x_t | x_{<t}) \tag{1}$$

This factorization relies on the autoregressive assumption, where each token depends solely on its preceding tokens. During training, the model is optimized by minimizing the negative log-likelihood loss over the dataset:

$$\mathcal{L} = -\frac{1}{T} \sum_{t=1}^{T} \log P(x_t | x_{<t}) \tag{2}$$

In autoregressive models, next-token prediction facilitates sequential generation by iteratively sampling tokens from the learned distribution $P(x_t | x_{<t})$. This approach is widely applicable multimodal domains such as visual understanding and visual generation.

### 3.2 Direct Preference Optimization

Direct Preference Optimization (DPO) provides a straightforward and efficient method by directly utilizing pairwise preference data to optimize the policy model. Specifically, given an input prompt $x$, and a preference data pair $(y_w, y_l)$, DPO aims to maximize the probability of the preferred output $y_w$ and minimize that of the undesirable output $y_l$. The optimization objective is formulated as:

$$\mathcal{L}_{\text{DPO}}(\theta) = -\mathbb{E}_{(x, y_w, y_l) \sim \mathcal{D}} \left[ \log \sigma \left( \beta \log \frac{\pi_\theta(y_w | x)}{\pi_{\text{ref}}(y_w | x)} - \beta \log \frac{\pi_\theta(y_l | x)}{\pi_{\text{ref}}(y_l | x)} \right) \right] \tag{3}$$

where $\mathcal{D}$ is the pair-wise preference dataset, $\sigma$ is the sigmoid function, $\pi_\theta(\cdot | x)$ is the policy model to be optimized, $\pi_{\text{ref}}(\cdot | x)$ is the reference model kept unchanged during training, and the hyperparameter $\beta$ controls the distance from the reference model.

## 4 Method

In this section, we present our method, HermesFlow, which curates pairwise preference data for both multimodal understanding and generation using homologous images and prompts, and seamlessly bridging the gap of multimodal understanding and generation through Pair-DPO training. An overview of HermesFlow is illustrated in Figure 3. In Section 4.1, we detail the methods for

curating homologous preference data for multimodal understanding and generation, respectively. In Section 4.2, we propose the Pair-DPO training strategy to bridge the gap between multimodal understanding and generation. In Section 4.3, we introduce self-play iterative optimization, enabling the self-improvement of MLLM over multiple iterations.

## 4.1 Curating Homologous Preference Data

**Homologous Input Data**    The curation of both multimodal understanding and generation preference data begins with homologous data $(x, y)$, where $y$ represents the caption or prompt of the image $x$.

**Understanding Preference Data**    We focus on the image captioning task to collect understanding preference data, which reflects the ability of MLLMs to capture visual features, including object attributes, spatial relationships, and detailed elements of both the subject and background. Give an image $x$, a pretrained MLLM is used to generate $n$ different captions according to the input prompt: *"Give a caption for this image."*. We then calculate the BERT similarity scores [2] $s(y, x)$ between the original prompt $y$ and each of the $n$ captions. The caption with the highest BERT similarity score is selected as the winning sample $y_w$, while the one with the lowest score is chosen as the losing sample $y_l$. Following this process, we construct the pairwise understanding preference data.

**Generation Preference Data**    Starting with the caption or prompt $y$, we use the pretrained MLLM to randomly generate $n$ images. Given that MLLM's understanding abilities surpass its generation capabilities, we apply a self-critique or self-selection method for choosing the generated data.

Specifically, given the prompt $y$, we use TIFA [14] to generate $q$ visual question-answer pairs, denoted as $\{(Q_1, A_1), (Q_2, A_2), \ldots, (Q_q, A_q)\}$. For each generated image, we evaluate them based on the accuracy of the VQA responses provided by the MLLM:

$$Acc(x_j) = \frac{1}{q} \sum_{i=1}^{q} \mathbb{I}(R_{j,i} = A_i), \ \ \forall j = 1, 2, \ldots, n \tag{4}$$

$$R_{j,i} = \text{MLLM}(x_j, Q_{j,i}) \tag{5}$$

where $R_{j,i}$ represents the response of MLLM according to the input of image $x_j$ and question $Q_{j,i}$. We select the image with the highest accuracy as the winning sample $x_w$ and the one with the lowest accuracy as the losing sample $x_l$, while also ensuring that the highest accuracy exceeds 0.6. Using this process, we construct the pairwise generation preference data.

**Homologous Output Preference Data**    After curating understanding and generation preference data from homologous input $(x, y)$ as mentioned above, where $y$ represents the caption or prompt of the image $x$, we obtain the homologous output preference data $\mathcal{D}$, denoted as $(x, y, x_w, x_l, y_w, y_l)$.

In this paper, we focus on exploring a preliminary method to achieve the self-improvement and self-alignment framework for understanding and generation in MLLMs, without the need for any external data as supervision. Although we curate preference data for understanding and generation using different methods, both are designed to capture the model's inherent biases accurately. These preference data do not represent two unrelated tasks, but rather offer two distinct perspectives on the same multimodal semantics.

## 4.2 Unified Enhancement with Pair-DPO

Homologous preference paired data of understanding and generation indicate the optimized directions for both capabilities of a pretrained MLLM within the same semantic space. To achieve joint optimization and alignment of understanding and generation, we introduce Pair-DPO. The optimization objective of Pair-DPO can be formulated as:

$$\mathcal{L}_{\text{Pair-DPO}}(\theta) = -\mathbb{E}_{(x,y,x_w,x_l,y_w,y_l) \sim \mathcal{D}} \left[ \log \sigma \left( \Delta_{Und} \Delta_{Gen} \right) \right] \tag{6}$$

$$\Delta_{Und} = \beta \log \frac{\pi_\theta(y_w \mid x)}{\pi_{\text{ref}}(y_w \mid x)} - \beta \log \frac{\pi_\theta(y_l \mid x)}{\pi_{\text{ref}}(y_l \mid x)} \tag{7}$$

$$\Delta_{Gen} = \beta \log \frac{\pi_\theta(x_w \mid y)}{\pi_{\text{ref}}(x_w \mid y)} - \beta \log \frac{\pi_\theta(x_l \mid y)}{\pi_{\text{ref}}(x_l \mid y)} \tag{8}$$

---

**Algorithm 1** The pseudocode of HermesFlow

---

**Input:** Homologous data $(x, y)$, pretrained model $\text{MLLM}_\theta$ with parameters $\theta$

1: **for** $i = 0, \dots, iter$ **do**
2:    **if** $i = 0$ **then**
3:       $y_w, y_l = \text{MLLM}_\theta^i(x)$   // Und preference data
4:       $x_w, x_l = \text{MLLM}_\theta^i(y)$   // Gen preference data
5:    **else**
6:       $y_1^i, y_2^i, \dots, y_n^i = \text{MLLM}_\theta^{i-1}(x)$
7:       $y_{\max}^i = \arg\max_{k \in \{1,\dots,n\}} \ s(y_k^i, x)$
8:       Update und-preference data using Equation (10)
9:       $x_1^i, x_2^i, \dots, x_n^i = \text{MLLM}_\theta^{i-1}(y)$
10:      $x_{\max}^i = \arg\max_{k \in \{1,\dots,n\}} \ Acc(x_k^i)$
11:      Update gen-preference data using Equation (10)
12:    **end if**
13:    Optimize $\text{MLLM}_\theta^{i-1}$ to $\text{MLLM}_\theta^i$ using Equation (6)
14: **end for**

---

where $\Delta_{Gen}$ and $\Delta_{Und}$ represent the preference differences in generation and understanding of MLLM, respectively. By using Pair-DPO to optimize homologous preference data jointly, we not only ensure mutual improvement in the understanding and generation capabilities of MLLM but also effectively narrow the gap between them. We provide the detailed derivation of the Pair-DPO optimization objective in Appendix A.

### 4.3 Self-Play Iterative Optimization

To achieve comprehensive optimization and achieve a convergence gap in understanding and generation of MLLMs, we introduce a novel yet easy self-play iterative optimization using Pair-DPO with multiple turns.

Take understanding preference data as an example. We denote the preference data curated in round $i - 1$ in Section 4.1 as $(y_w^{i-1}, y_l^{i-1})$. In the optimization of round $i$, the optimized MLLM generates $n$ new captions $(y_1^i, y_2^i, \dots, y_n^i)$ from the input of image $x$. The preference data is selected based on the following rules:

$$y_{\max}^i = \arg \max_{k \in \{1,\dots,n\}} \ s(y_k^i, y) \tag{9}$$

$$(y_w^i, y_l^i) = \begin{cases} (y_{\max}^i, y_w^{i-1}) & \text{if } s(y_{\max}^i, y) > s(y_w^{i-1}, y) \\ (y_{\max}^i, y_l^{i-1}) & \text{otherwise} \end{cases} \tag{10}$$

where $s(y_k^i, y)$ denotes the BERT similarity score between the generated caption $y_k^i$ and the homologous input caption $y$. Select the caption $y_{\max}^i$ with the highest similarity score, which represents the local upper bound of the optimized MLLM's understanding capability. If $s(y_{\max}^i, y) > s(y_w^{i-1}, y)$, MLLM has effectively learned preference knowledge from the previous round. Therefore, it needs to be updated and further optimized using the higher-quality sample $y_{\max}^i$ as the benchmark. Conversely, if $s(y_{\max}^i, y) < s(y_w^{i-1}, y)$, effective optimization was not achieved in the previous round. In this case, it is necessary to update with simpler and clearer preference data $y_{\max}^i$ as the winning sample to provide a smoother learning gradient. Through iterative optimization, we achieve self-improvement of MLLM without relying on any external high-quality training data.

## 5 Experiments

### 5.1 Experimental Setup

**Training Setup** We randomly select 5,000 image-caption pairs from JourneyDB [33] as our homologous input data. For the Visual Question Answering (VQA) data corresponding to each pair, we combine the VQA from JourneyDB with the VQA generated from TIFA [14] for a comprehensive evaluation. Our HermesFlow is trained upon Show-o [48], using a batch size of 4 for both caption and generation data over 3,000 steps. We employ the AdamW optimizer with a weight decay of 0.01,

Table 1: Evaluation on multimodal understanding benchmarks. The baseline data is quoted from Show-o [48].

| Model | # Params | POPE↑ | MME↑ | Flickr30k↑ | VQAv2$_{(test)}$↑ | GQA↑ | MMMU↑ |
|---|---|---|---|---|---|---|---|
| Gemini-Nano-1 [37] | 1.8B | - | - | - | 62.7 | - | 26.3 |
| Emu [35] | 13B | - | - | 77.4 | 57.2 | - | - |
| NExT-GPT [46] | 13B | - | - | 84.5 | 66.7 | - | - |
| SEED-X [6] | 17B | 84.2 | 1435.7 | 52.3 | - | 47.9 | 35.6 |
| Chameleon [36] | 34B | - | - | 74.7 | 66.0 | - | - |
| Show-o [48] | 1.3B | 80.0 | 1232.9 | 67.6 | 74.7 | 61.0 | 27.4 |
| **HermesFlow (Ours)** | 1.3B | 81.4 | 1249.7 | 69.2 | 75.3 | 61.7 | 28.3 |

Table 2: Evaluation on visual generation benchmarks: GenEval [7] and DPG-Bench [13].

| Methods | #params | GenEval↑ | | | | | | | DPG-Bench↑ |
|---|---|---|---|---|---|---|---|---|---|
| | | Single Obj. | Two Obj. | Counting | Colors | Position | Color Attri. | Overall | Average |
| *Diffusion Model* | | | | | | | | | |
| LDM [30] | 1.4B | 0.92 | 0.29 | 0.23 | 0.70 | 0.02 | 0.05 | 0.37 | - |
| DALL-E 2 [29] | 4.2B | 0.94 | 0.66 | 0.49 | 0.77 | 0.10 | 0.19 | 0.52 | - |
| SD 1.5 [30] | 860M | 0.94 | 0.37 | 0.27 | 0.72 | 0.05 | 0.07 | 0.40 | 63.18 |
| SD 2.1 [30] | 865M | 0.97 | 0.50 | 0.46 | 0.80 | 0.07 | 0.14 | 0.49 | 68.09 |
| *Autoregressive Model* | | | | | | | | | |
| LlamaGen [34] | 775M | 0.87 | 0.25 | 0.23 | 0.51 | 0.06 | 0.04 | 0.32 | 65.16 |
| Emu [35] | 14B | 0.87 | 0.34 | 0.26 | 0.56 | 0.07 | 0.06 | 0.36 | - |
| Chameleon [36] | 34B | 0.89 | 0.39 | 0.28 | 0.66 | 0.08 | 0.07 | 0.40 | - |
| LWM [21] | 7B | 0.93 | 0.41 | 0.46 | 0.79 | 0.09 | 0.15 | 0.47 | - |
| SEED-X [6] | 17B | 0.97 | 0.58 | 0.26 | 0.80 | 0.19 | 0.14 | 0.49 | - |
| Show-o [48] | 1.3B | 0.98 | 0.77 | 0.58 | 0.81 | 0.23 | 0.44 | 0.64 | 67.48 |
| Janus [44] | 1.3B | 0.97 | 0.68 | 0.30 | **0.84** | **0.46** | 0.42 | 0.61 | - |
| **HermesFlow (Ours)** | 1.3B | **0.98** | **0.84** | **0.66** | 0.82 | 0.32 | **0.52** | **0.69** | **70.22** |

and an initial learning rate of 2e-5 with a cosine scheduling. The parameter $\beta$ for Pair-DPO is set to 0.2. All experiments are conducted under 8*NVIDIA A100 GPUs.

Notably, while we use JourneyDB's prompts and images solely to construct preference data, we do not treat them as direct supervision targets. Instead, our supervision stems from preference pairs generated by the model itself, making HermesFlow a self-improvement framework for MLLMs. The JourneyDB data merely serves as model input, while training and alignment are driven by the model's own output quality. This design enables the framework to work with arbitrary image-prompt pairs, enhancing its flexibility and applicability.

**Evaluation Metrics**    To assess multimodal understanding capabilities, we evaluate using POPE [20], MME [5], Flickr30k [25], VQAv2 [8], GQA [15], and MMMU [58]. For visual generation capabilities, we use GenEval [7] and DPG-Bench [13] to evaluate the model's prompt-image alignment. We further assess image fidelity with FID [12] and CLIP-Score [27]. Additionally, we conduct a comprehensive user study to objectively compare our model with other baselines.

## 5.2   Main Results

**Multimodal Understanding Performances**    Table 1 summarizes the comparison between our method and other leading MLLMs on multimodal understanding benchmarks. Notably, HermesFlow achieves similar or superior understanding performance compared to larger models like SEED-X and Chameleon, using less than 1/10 of the parameters. Additionally, HermesFlow demonstrates significant strengths across all metrics compared to Show-o, indicating that Pair-DPO effectively reduces the understanding-generation gap while maintaining or even enhancing understanding ability. For more qualitative examples on multimodal understanding, please refer to Section D.

**Image Generation Performances**    As shown in Figure 4, HermesFlow achieves superior generation results compared to three powerful Multimodal LLMs: VILA-U [47], Janus [44], and Show-o [48]. Compared to its backbone, Show-o, HermesFlow demonstrates superior performance in generating object attributes and accurate counting. This improvement stems from its stronger understanding

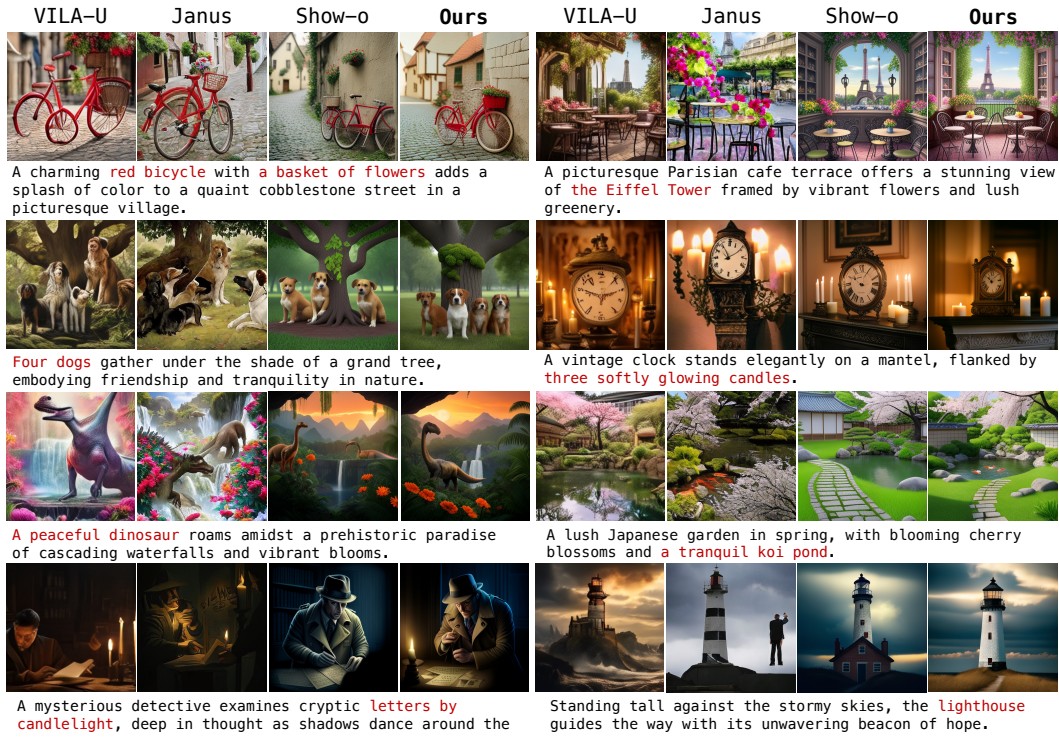

| VILA-U | Janus | Show-o | **Ours** | VILA-U | Janus | Show-o | **Ours** |

A charming red bicycle with a basket of flowers adds a splash of color to a quaint cobblestone street in a picturesque village.

A picturesque Parisian cafe terrace offers a stunning view of the Eiffel Tower framed by vibrant flowers and lush greenery.

Four dogs gather under the shade of a grand tree, embodying friendship and tranquility in nature.

A vintage clock stands elegantly on a mantel, flanked by three softly glowing candles.

A peaceful dinosaur roams amidst a prehistoric paradise of cascading waterfalls and vibrant blooms.

A lush Japanese garden in spring, with blooming cherry blossoms and a tranquil koi pond.

A mysterious detective examines cryptic letters by candlelight, deep in thought as shadows dance around the room.

Standing tall against the stormy skies, the lighthouse guides the way with its unwavering beacon of hope.

Figure 4: Qualitative comparison between our HermesFlow and three outstanding Multimodal LLMs VILA-U [47], Janus [44], and Show-o [48]. Colored text denotes the advantages of HermesFlow in generated images.

Table 3: MSCOCO zero-shot FID and CLIP-Score.

| Method | # Params | FID↓ | CLIP-Score↑ |
|---|---|---|---|
| LDM [30] | 1.4B | 12.64 | - |
| DALL·E 2 [29] | 6.5B | 10.39 | - |
| SD 1.5 [30] | 860M | 9.62 | 30.23 |
| SD 2.1 [30] | 865M | 8.03 | 30.87 |
| LlamaGen [34] | 775M | 9.45 | 29.12 |
| Emu [35] | 14B | 11.02 | 28.98 |
| LWM [21] | 7B | 12.68 | - |
| SEED-X [6] | 17B | 14.99 | - |
| Show-o [48] | 1.3B | 9.24 | 30.63 |
| **HermesFlow (Ours)** | 1.3B | **9.07** | **31.08** |

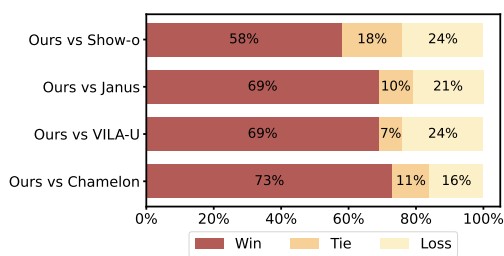

Figure 5: Results of user study.

capabilities, which are utilized to filter generated images and achieve mutual refinement through Pair-DPO iteratively.

We compare HermesFlow with other visual generation models on GenEval [7] and DPG-Bench [13], as shown in Table 2. Compared to the diffusion-based generative model SD 2.1 [30], HermesFlow demonstrates remarkable performance across all benchmarks. Furthermore, it surpasses larger autoregressive models, such as Chameleon [36] and LWM [21]. When compared to Show-o [48], HermesFlow exhibits significant strengths in object counting and positions, this is attributed to the critique of its superior understanding capability, which greatly enhances its visual generation performance in aspects such as object quantity, location, and attributes. We present the zero-shot FID [12] and CLIP-Score [27] of HermesFlow on MSCOCO-30K in Table 3. The results clearly show that after the iterative optimization with Pair-DPO, HermesFlow achieves improved performance in both image fidelity and prompt-image alignment.

We also conducted a comprehensive user study to evaluate the effectiveness of HermesFlow in visual generation. As illustrated in Figure 5, we randomly selected 25 prompts for each comparison, and invited 35 users from diverse backgrounds to vote on image generation quality, collecting a total of 3,500 votes. Alignment between the generated images and the prompts was used as the

Table 4: Quantitative assess of MLLM's Understanding and Generation Gap.

| Method | # Params | Understanding Score↑ | Generation Score ↑ | Gap ↓ |
|---|---|---|---|---|
| VILA-U [47] [48] | 7B | 0.646 | 0.477 | 0.169 |
| Janus [44] | 1.3B | 0.599 | 0.417 | 0.182 |
| Show-o [48] | 1.3B | 0.520 | 0.433 | 0.087 |
| **HermesFlow (Ours)** | 1.3B | 0.533 | 0.497 | **0.036** |

Table 5: Comparison of Pair-DPO vs. DPO and the Effect of Pair-DPO Iterations.

| Methods | Understanding Bench | | | Generation Bench | |
|---|---|---|---|---|---|
| | POPE↑ | MME↑ | MMMU↑ | GenEval (Overall)↑ | DPG-Bench (Average)↑ |
| Show-o [48] | 80.0 | 1232.9 | 27.4 | 0.64 | 67.48 |
| DPO (Understanding) | 80.8 | 1242.2 | 27.8 | 0.58 | 67.88 |
| DPO (Generation) | 80.5 | 1239.3 | 27.5 | 0.70 | 70.03 |
| Pair-DPO (Iter. 0) (Show-o) | 80.0 | 1232.9 | 27.4 | 0.64 | 67.48 |
| Pair-DPO (Iter. 1) | 81.1 | 1246.7 | 28.0 | 0.68 | 70.19 |
| Pair-DPO (Iter. 2) | 81.3 | 1248.3 | 28.1 | 0.69 | 70.21 |
| Pair-DPO (Iter. 3) | **81.4** | **1249.7** | **28.3** | **0.69** | **70.22** |

primary evaluation criterion, with aesthetic quality and detail completeness considered under the same conditions. The results demonstrate that HermesFlow received widespread user approval in visual generation.

**Quantitative assess of MLLM's Understanding and Generation Gap**  As shown in Figure 2, we use homologous data consisting of caption/prompt $y$ and image $x$ as input to evaluate the capability of understanding and generation respectively. The homologous data is randomly selected from JourneyDB [33]. For the understanding task, to ensure comprehensive and high-quality question-answer (QA) pairs, we first use TIFA [14] to generate QA pairs based on the image and caption. These are then augmented with QA pairs from JourneyDB to create a more thorough and in-depth dataset. The final understanding score is calculated as the average accuracy of the answers. For the generation task, we use the prompt as input to generate an image for each prompt. These generated images are evaluated by posing the same set of questions to GPT-4o [16], with the final generation score determined by the average accuracy of GPT-4o's answers. Since the generation capabilities of MLLMs are relatively limited, strict evaluation criteria are applied in cases of severe object blurring or significant loss of details. Therefore, GPT-4o is required to carefully analyze the completeness and authenticity of the objects involved in each question before providing answers. This evaluation pipeline was applied to multiple MLLMs, with the results presented in Table 4.

It is clear that a significant gap exists between multimodal understanding and generation in MLLM. HermesFlow seamlessly bridges this gap through self-play iterative optimization using Pair-DPO from homologous preference data.

## 5.3 Ablation Study

**Pair-DPO vs. DPO**  Pair-DPO can simultaneously enhance both the understanding and generation capabilities of multimodal LLMs. As shown in Table 5, compared to DPO methods that rely solely on understanding or generation preference, a single round of Pair-DPO achieves superior performance by jointly optimizing both capabilities through the use of multimodal preference data. Furthermore, we observed that when using preference data from only one modality, whether understanding or generation, the capability of the other modality also improves. This demonstrates the same findings in MetaMorph [40] and Liquid [45] that multimodal understanding and generation are synergistic.

**Self-play Iterative Optimization**  As shown in Table 5, we conducted an experimental analysis to examine the impact of iterations in self-play iterative optimization. It is evident that the first round of iterative optimization yields the most significant improvements in both understanding and generation. This is because the notable gap between the understanding and generation capabilities of MLLMs is most effectively bridged in the initial iteration. When the number of iterations exceeds 2, we

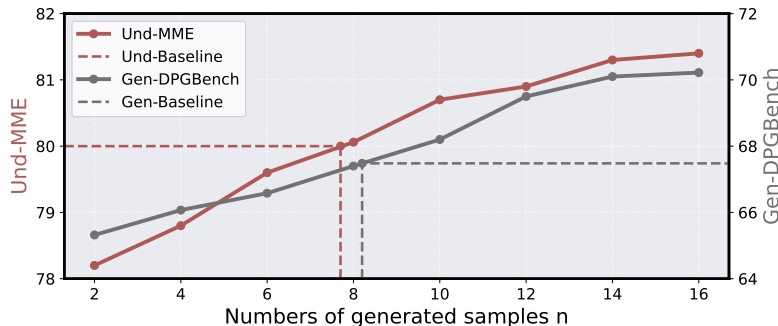

Figure 6: Influence of the richness of each preference sample.

observed that understanding ability continues to improve slightly, while generation ability remains almost stable. We argue that since generation is a fine-grained visual task, cross-capability transfer has limited impact on further enhancing generation ability in subsequent iterations.

**The Impact of Each Preference Sample Richness**   The performance of Pair-DPO is largely influenced by the number of generated samples $n$ for both understanding and generation data. We conducted experiments to analyze the impact of $n$ on both understanding and generation in MLLMs, with results shown in Figure 6. The dashed lines represent the performance of the baseline model, Show-o [48].

When $n$ is too small, the model's understanding and generation performance decline. This is due to the insufficient number of samples and the limited capability of the baseline model, which leads to a noisy preference dataset and significantly impacts the results. However, as the sample size increases, it enables more accurate identification of the model's optimal local upper bound, which in turn facilitates the curation of higher-quality preference data, leading to noticeable improvements in the understanding and generation capabilities of MLLMs.

Furthermore, Figure 6 reveals that achieving performance comparable to the baseline in generation requires more sampling data that understanding. This indicates that the generation capabilities of MLLMs are more sensitive to noise in the preference data, highlighting a greater need for high-quality generation data.

## 6   Conclusion

In this paper, we present a new MLLM alignment paradigm, HermesFlow, to seamlessly bridge the gap between multimodal understanding and generation. By iterative optimized with Pair-DPO using homologous preference data, HermesFlow successfully Improve the capabilities of both multimodal understanding and generation while narrowing the gap between them. Our extensive empirical evaluations across diverse understanding and generation benchmarks demonstrate the effectiveness of HermesFlow. However, due to current limitations in the number and capabilities of open-source MLLMs, HermesFlow has not yet been optimized across a wider range of backbones. HermesFlow has the potential as a general alignment framework for next-generation multimodal foundation models. In the future, we plan to extend this framework: (i). Extending HermesFlow to diffusion foundation models like MMaDA [52] to evaluate its generalization and performance on various backbones; (ii) Combining with more advanced RL algorithms [42, 67, 43] to enhance multimodal inteligence for different scenarios.

## Acknowledgement

This work is supported by National Natural Science Foundation of China (U22B2037), Beijing Municipal Science and Technology Project (Z231100010323002), research grant No. SH-2024JK29, PKU-Tencent joint research Lab, and High-performance Computing Platform of Peking University.

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

# A Derivation of the Pair-DPO Optimization Objective

Considering that the optimization objective of standard Direct Preference Optimization is:

$$\mathcal{L}_{\text{DPO}}(\theta) = -\mathbb{E}_{(x,y_w,y_l)\sim\mathcal{D}}\left[\log\sigma\left(\beta\log\frac{\pi_\theta(y_w\,|\,x)}{\pi_{\text{ref}}(y_w\,|\,x)}-\beta\log\frac{\pi_\theta(y_l\,|\,x)}{\pi_{\text{ref}}(y_l\,|\,x)}\right)\right] \tag{11}$$

Pair-DPO simultaneously optimizes understanding and generation using pairedcpreference data, with its loss function comprising these two components:

$$\mathcal{L}_{\text{Pair-DPO}}(\theta) = \mathcal{L}_{Und}(\theta) + \mathcal{L}_{Gen}(\theta)$$
$$= -\mathbb{E}_{(x,y,x_w,x_l,y_w,y_l)\sim\mathcal{D}}$$
$$\left[\log\sigma\left(\beta\log\frac{\pi_\theta(y_w\,|\,x)}{\pi_{\text{ref}}(y_w\,|\,x)}-\beta\log\frac{\pi_\theta(y_l\,|\,x)}{\pi_{\text{ref}}(y_l\,|\,x)}\right)+\log\sigma\left(\beta\log\frac{\pi_\theta(x_w\,|\,y)}{\pi_{\text{ref}}(x_w\,|\,y)}-\beta\log\frac{\pi_\theta(x_l\,|\,y)}{\pi_{\text{ref}}(x_l\,|\,y)}\right)\right]$$
$$= -\mathbb{E}_{(x,y,x_w,x_l,y_w,y_l)\sim\mathcal{D}}$$
$$\left[\log\sigma\left(\beta\log\frac{\pi_\theta(y_w\,|\,x)}{\pi_{\text{ref}}(y_w\,|\,x)}-\beta\log\frac{\pi_\theta(y_l\,|\,x)}{\pi_{\text{ref}}(y_l\,|\,x)}\right)\left(\beta\log\frac{\pi_\theta(x_w\,|\,y)}{\pi_{\text{ref}}(x_w\,|\,y)}-\beta\log\frac{\pi_\theta(x_l\,|\,y)}{\pi_{\text{ref}}(x_l\,|\,y)}\right)\right] \tag{12}$$

Here, $\Delta_{Und}$ and $\Delta_{Gen}$ are defined as:

$$\Delta_{Und} = \beta\log\frac{\pi_\theta(y_w\,|\,x)}{\pi_{\text{ref}}(y_w\,|\,x)} - \beta\log\frac{\pi_\theta(y_l\,|\,x)}{\pi_{\text{ref}}(y_l\,|\,x)} \tag{13}$$

$$\Delta_{Gen} = \beta\log\frac{\pi_\theta(x_w\,|\,y)}{\pi_{\text{ref}}(x_w\,|\,y)} - \beta\log\frac{\pi_\theta(x_l\,|\,y)}{\pi_{\text{ref}}(x_l\,|\,y)} \tag{14}$$

Substituting these definitions, the final Pair-DPO objective can be expressed as:

$$\mathcal{L}_{\text{Pair-DPO}}(\theta) = -\mathbb{E}_{(x,y,x_w,x_l,y_w,y_l)\sim\mathcal{D}}\left[\log\sigma\left(\Delta_{Und}\Delta_{Gen}\right)\right] \tag{15}$$

# B More Quantitative Results

Table 6: Evaluation on MME-Unify [50].

| Methods | #params | Understanding | | | | Generation | MME-U Score |
| | | SIPU | MITIU | VPU | Avg | TIG | Avg |
| --- | --- | --- | --- | --- | --- | --- | --- |
| GILL [17] | 7B | 22.18 | 6.00 | 3.56 | 10.58 | 46.60 | 6.12 |
| MiniGPT-5 [68] | 7B | 19.25 | 10.92 | 15.93 | 15.37 | 35.48 | 7.09 |
| Show-o [48] | 1.3B | 32.47 | 34.75 | 25.66 | 30.96 | 43.54 | 12.74 |
| Emu3 [41] | 8B | 45.75 | 30.50 | 23.32 | 33.19 | 49.08 | 13.79 |
| **HermesFlow (Ours)** | 1.3B | 41.49 | 33.00 | 28.32 | 34.27 | 46.48 | 14.01 |

To more comprehensively evaluate HermesFlow's capabilities in both understanding and generation, we present results on the more extensive MME-Unify [50] benchmark. As shown in Table 6, with the backbone Show-o, HermesFlow achieves significant improvements in both understanding and generation, indicating that our approach not only reduces the gap between the two but also facilitates their joint enhancement. Despite using fewer parameters (1.3B), HermesFlow outperforms larger models such as Emu3 with 7B parameters, further demonstrating its potential as a general post-training framework for MLLMs.

# C   Example of Paired Preference Data

**Homologous Input Data:**

**Prompt/Caption:**
Two samurais, one black and the other white, fiercely engage in combat as the black samurai skillfully blocks an incoming sword attack amidst a city backdrop, creating a scene of intense and tense action.

**Image:**

**Sampled data (n=10):**    Win  Lose

**Sampled Captions for Understanding:**
**Caption1:** Two warriors in armor stand in front of a city, holding swords.
**Caption2:** Two warriors in armor, one with a sword and the other with a spear, are standing in front of a city.
**Caption3:** A character in the game Shadowrun stands on a platform with two other characters.
**Caption4:** A scene of two warriors standing in front of a city. One warrior has a sword in each hand, and they are facing each other.
**Caption5:** Two warriors in armor, one with a sword and the other with a spear, are standing in front of a city.
**Caption6:** Two knights in armor are standing in front of a city.
**Caption7:** The two knights are in a dark city, standing on a street with swords in hand. They seem to have just come out from the shadows, ready for battle.
**Caption8:** Two warriors in armor, one with a sword and the other with a spear, are standing in front of a city.
**Caption9:** Two warriors in armor stand on a platform, holding swords. One of the warriors has a horned helmet, the other has a helmet with horns, city background.
**Caption10:** The image shows two knights on the edge of a cliff. They have two swords and are facing off, preparing to fight.

**Sampled Images for Generation:**

| Image1 | Image2 | Image3 | Image4 | Image5 |
| Image6 | Image7 | Image8 | Image9 | Image10 |

**Paired Preference Data:**

**Understanding:**
**Caption_win:** Two warriors in armor stand on a platform, holding swords. One of the warriors has a horned helmet, the other has a helmet with horns, city background.
**Caption_lose:** A character in the game Shadowrun stands on a platform with two other characters.

**Generation:**

**Image_win:**        **Image_lose:**

Figure 7: An example of the curation of paired preference data.

# D    Qualitative Examples on Multimodal Understanding

---

**Qualitative Example of Multimodal Understanding (1)**

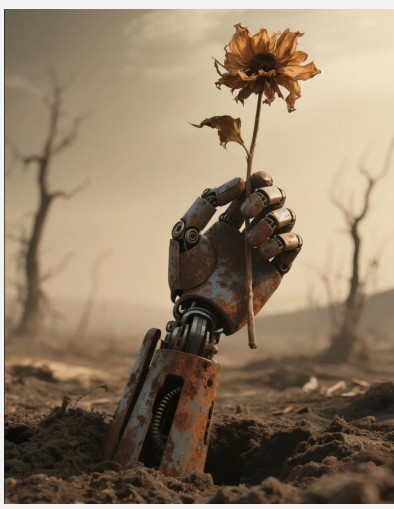

**HermesFlow**:
The image features a rusty, old-fashioned, and somewhat eerie-looking metal sculpture of a hand holding a flower. The sculpture is placed on a patch of dirt, surrounded by a barren landscape. The hand appears to be holding a flower, which is a single, wilted flower. The overall scene gives off a sense of abandonment and decay, as the sculpture seems to be a relic from a bygone era.

**Prompt:** Please describe this image in detail.

---

**Qualitative Example of Multimodal Understanding (2)**

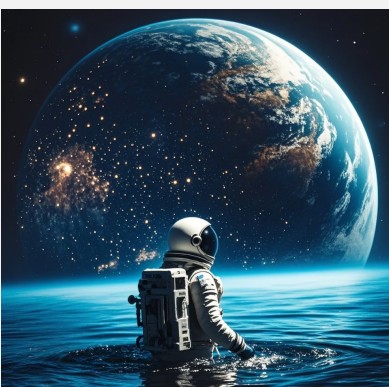

**HermesFlow**:
The image features an astronaut floating in the ocean, with a large planet visible in the background. The astronaut is wearing a spacesuit and appears to be floating on the surface of the water. The scene is set in outer space, with a star-filled sky visible in the background. The astronaut seems to be gazing at the planet, possibly contemplating their journey or the vastness of the universe.

**Prompt:** Please describe this image in detail.

## Qualitative Example of Multimodal Understanding (3)

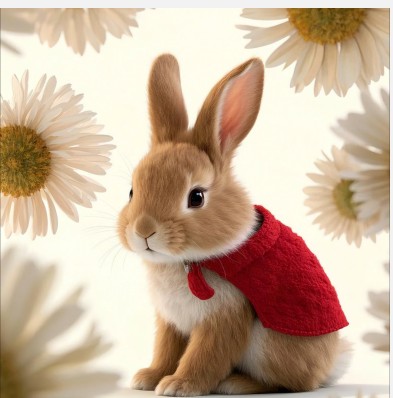

**HermesFlow**:
The image features a small, adorable bunny wearing a red sweater, sitting on a white surface. The bunny is surrounded by beautiful arrangement of white daisies, with some of the flowers positioned behind the bunny, creating a visually pleasing scene. The bunny appears to be the main focus of the image, with its cute appearance and the contrasting red sweater standing out against the white daisies.

**Prompt:** Please describe this image in detail.

## Qualitative Example of Multimodal Understanding (4)

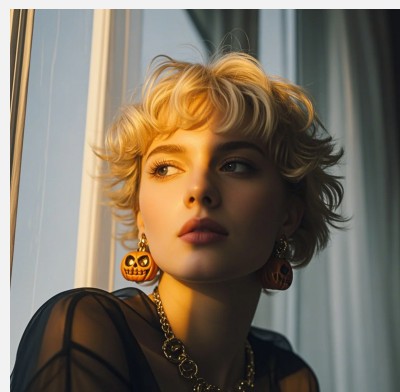

**HermesFlow**:
The image features a woman with blonde hair, wearing a black dress, and looking out of a window. She is positioned in the center of the frame, with her gaze directed towards something outside the window. The woman is wearing large earrings, which are noticeable as she looks out the window. The scene appears to be set indoors, with the woman standing near a window, possibly in a room.

**Prompt:** Please describe this image in detail.

