# OpenReview forum: "HermesFlow: Seamlessly Closing the Gap in Multimodal Understanding and Generation"
_NeurIPS.cc/2025/Conference — NeurIPS 2025 poster_

### Official Review · Reviewer_zMCR · 2025-06-30

**Clarity:** 3
**Significance:** 3
**Originality:** 3
**Rating:** 4
**Confidence:** 4

**Summary:**

This paper introduces HermesFlow, a framework intended to bridge the notable gap between multimodal understanding and generation capabilities in unified Multimodal Large Language Models (MLLMs). They propose a methodology called Pair-DPO, leveraging paired preference data for joint optimization, which is iteratively refined via a self-play strategy. The authors demonstrate that their approach, built on the Show-o model, narrows the capability gap while improving performance in both domains.

**Questions:**

1. Can you do more experiments on other backbone models, like Janus?
2. Given the dependency on homologous data quality, could you provide more details about the methods or heuristics you used to ensure diversity and quality in the homologous preference dataset?
3. What are the framework's failure modes if a base model is too weak to generate high-quality candidate samples for the preference curation process?
4. Janus and VILA-U are used as baselines for generation in Figure 4 and for gap analysis in Table 4. Why are their quantitative results on the standard understanding benchmarks in Table 1 not included?

**Ethical Concerns:**

["NO or VERY MINOR ethics concerns only"]

**Final Justification:**

The authors provide experiments about generalization and data quality. Therefore, I keep my positive scores.

**Limitations:**

Authors have discussed their limitations.

**Paper Formatting Concerns:**

No Paper Formatting Concerns

**Quality:**

3

**Strengths And Weaknesses:**

Strengths:
1. The paper precisely identifies a significant issue within unified MLLMs, where the capability between understanding and generation is imbalanced.
2. HermesFlow leverages Pair-DPO, a creative approach employing homologous preference data to align multimodal understanding and generation effectively. The self-play iterative optimization scheme is particularly interesting, allowing the model to improve autonomously without external data supervision.
3. Extensive experiments and comparisons with other baselines demonstrate HermesFlow's superiority.

Weaknesses:
1. Limited Generalization Validation. The claim that HermesFlow is a "general alignment framework"  is not sufficiently supported by the experiments. The entire validation is performed on a single backbone model, Show-o. While the results are promising, it is impossible to know if they generalize to other MLLMs with different architectures (e.g., Emu3, Janus, VILA-U).
2. Homologous Data Dependency. The proposed method’s efficacy strongly relies on generating high-quality homologous data pairs. However, the quality and diversity of these pairs greatly influence performance, as briefly mentioned in ablation studies.
3. Computational Efficiency and Scalability Concerns. The iterative self-play optimization might impose significant computational costs. The paper does not sufficiently discuss potential efficiency issues or computational complexity.

---

> ### Author Rebuttal · Authors · 2025-07-31
>
> *We sincerely thank you for your time and efforts in reviewing our paper and for your valuable feedback. We are glad to see that the topic of our paper is significant and important, the proposed method is interesting, and the experiments are extensive. Please see below for our responses to your comments.*
>
> >**Q1: Limited Generalization Validation. While the results are promising, it is impossible to know if they generalize to other MLLMs with different architectures.**
>
> **A1:** We apologize for the oversight. Here, **we reproduced HermesFlow on both Janus (1.3B) and VILA-U (7B)**. The results are shown below:
>
> **Table 1: Evaluation on Multimodal Understanding Benchmarks.**
>
> |                        | #Params | POPE     | MME        | Flickr30k | VQAv2    | GQA      | MMMU     |
> | ---------------------- | ------- | -------- | ---------- | --------- | -------- | -------- | -------- |
> | Show-o                 | 1.3B    | 80.0     | 1232.9     | 67.6      | 74.7     | 61.0     | 27.4     |
> | **HermesFlow(Show-o)** | 1.3B    | **81.4** | **1249.7** | **69.2**  | **75.3** | **61.7** | **28.3** |
> | Janus                  | 1.3B    | 87.0     | 1338.0     | 68.8      | 77.3     | 59.1     | 30.5     |
> | **HermesFlow(Janus)**  | 1.3B    | **87.4** | **1377.3** | **70.2**  | **78.4** | **60.0** | **30.9** |
> | VILA-U                 | 7B      | 85.8     | 1401.8     | 69.9      | 79.4     | 60.8     | 30.1     |
> | **HermesFlow(VILA-U)** | 7B      | **86.9** | **1427.1** | **70.8**  | **79.7** | **61.4** | **31.8** |
>
> **Table 2: Evaluation on Visual Generation Benchmark: GenEval.**
>
> |                        | #Params | Single Obj. | Two Obj. | Counting | Colors   | Position | Color Attri. | Overall  |
> | ---------------------- | ------- | ----------- | -------- | -------- | -------- | -------- | ------------ | -------- |
> | Show-o                 | 1.3B    | 0.98        | 0.77     | 0.58     | 0.81     | 0.23     | 0.44         | 0.64     |
> | **HermesFlow(Show-o)** | 1.3B    | **0.98**    | **0.84** | **0.66** | **0.82** | **0.32** | **0.52**     | **0.69** |
> | Janus                  | 1.3B    | 0.97        | 0.68     | 0.30     | 0.84     | 0.46     | 0.42         | 0.61     |
> | **HermesFlow(Janus)**  | 1.3B    | **0.98**    | **0.89** | **0.48** | **0.87** | **0.56** | **0.55**     | **0.72** |
> | VILA-U                 | 7B      | 0.95        | 0.62     | 0.48     | 0.84     | 0.16     | 0.36         | 0.57     |
> | **HermesFlow(VILA-U)** | 7B      | **0.98**    | **0.88** | **0.65** | **0.85** | **0.39** | **0.53**     | **0.71** |
>
> HermesFlow still shows significant improvements on backbones of larger sizes and different architectures. It can be observed from the table that HermesFlow demonstrates remarkable enhancements in Two Obj., Counting, Position, and Color Attribute on both the 1.3B and 7B backbones. This is because of the excellent understanding ability of MLLM regarding image attributes, spatial position relationships, etc., which makes the preference data contain a large number of features in this regard. This indicates that our architecture has **strong generalization ability and is a general unified model post-training framework**. We will incorporate the relevant experimental updates in the revised manuscript. We appreciate your valuable feedback.
>
>
>
> >**Q2: Homologous Data Dependency. The proposed method’s efficacy strongly relies on generating high-quality homologous data pairs. However, the quality and diversity of these pairs greatly influence performance, as briefly mentioned in ablation studies.**
>
>  **A2:** We thank the reviewer for raising this important point. The quality of homologous data pairs has a significant impact on the final training results. What we explore here is a **fully self-improvement and self-alignment paradigm that requires no external supervision**. We find that simply leveraging the gap between the MLLM’s own understanding and generation capabilities is sufficient to effectively align the two and achieve joint improvement.
>
> However, constructing higher-quality preference data is indeed a highly valuable and promising direction. Moving beyond the framework limitations of model self-improvement or self-critique, we use **Gemini-2.5-Pro as a judge** to collect preference datasets from multiple generated data samples.:
>
> **Table 3: Use Gemini-2.5-Pro as the Judge to Create Preference Data**
>
> |                               | #Params | POPE | MME    | MMMU | GenEval (Overall) | DPG-Bench (Average) |
> | ----------------------------- | ------- | ---- | ------ | ---- | ---------------- | ------------------- |
> | Show-o                        | 1.3B    | 80.0 | 1232.9 | 27.4 | 0.64             | 67.48               |
> | HermesFlow (Self-improvement) | 1.3B    | 81.4 | 1249.7 | 28.3 | 0.69             | 70.22               |
> | HermesFlow (MLLM-as-a-judge)  | 1.3B    | 81.7 | 1258.4 | 28.7 | 0.73             | 70.99               |
>
> From the table, employing a more powerful external model as a data collector yields better results in both understanding and generation. This is because Gemini-2.5-Pro is capable of thoroughly mining and selecting the optimal win and lose samples, which also highlights the potential of the HermesFlow architecture. We will investigate better data construction approaches to further enhance MLLM performance and close existing gaps.
>
> >**Q3: Computational Efficiency and Scalability Concerns. The iterative self-play optimization might impose significant computational costs. The paper does not sufficiently discuss potential efficiency issues or computational complexity.**
>
> **A3:** Thank you for the constructive feedback. Pair-DPO maintains comparable computational requirements to standard DPO. Although it introduces homologous preference data construction and iterative optimization, the small dataset size (5k image-text pairs) and low overhead per iteration (typically 2-3 rounds) prevent significant resource increases. In practice, training can be efficiently performed using just 4×A100 GPUs on Show-o.
>
> Here is an approximate listing of the training time for each stage (all our experiments were conducted on 4*NVIDIA A100-80G GPUs):
>
> **Table 4: The Training Time for Each Stage.**
> | Phrase      | Training Time |
> | :---------- | :------------ |
> | Iteration 1 | ~4h 12min     |
> | Iteration2  | ~1h 46min     |
> | Iteration3  | ~1h 46min     |
> | Total       | ~7h 44min     |
>
> To mitigate overfitting, we used fewer training samples and fewer training steps starting from iteration 2. As a result, with the continuous refinement of iterative feedback learning, the training time of the model decreases progressively.
>
> >**Q4: Can you do more experiments on other backbone models, like Janus?**
>
> **A4:** Thank you for your suggestion. We provide the results on Janus and VILA-U **in Tables 1 and 2 in A1.**
>
> >**Q5: Given the dependency on homologous data quality, could you provide more details about the methods or heuristics you used to ensure diversity and quality in the homologous preference dataset?**
>
> **A5:** We thank the reviewer for raising this important point. We sample the homologous data from JourneyDB, a comprehensive and high-quality dataset. During the filtering process, we apply a **strict threshold of 0.8 using BERT similarity score** to control the diversity of the prompts in the homologous data. Additionally, we use the **CLIP score to filter prompt-image pairs, with a threshold set at 0.3** to ensure the quality of text-image pairs.
>
> >**Q6: What are the framework's failure modes if a base model is too weak to generate high-quality candidate samples for the preference curation process?**
>
> **A6:** Thank you for this insightful question. HermesFlow is a highly flexible architecture that enables parameter tuning to improve the quality of homologous preference data. For weaker backbones, we **increase the number of samples per preference pair from 10 to 16**, which leads to a notable improvement in the quality of the collected preferences. Additionally, an auxiliary judge model can be introduced to further support preference data collection under weaker backbone settings. Our experimental results on **Chameleon-7B** are presented below:
>
> **Table 5: Results of Hermesflow Using a Weaker Backbone.**
> |                        | #Params | POPE | MME   | MMMU | GenEval(Overall) | DPG-Bench (Average) |
> | ---------------------- | ------- | ---- | ----- | ---- | ---------------- | ------------------- |
> | Chameleon              | 7B      | 72.1 | 947.7 | 26.2 | 0.39             | 66.10               |
> | HermesFlow (Chameleon) | 7B      | 72.2 | 968.1 | 26.4 | 0.44             | 68.94               |
>
> Increasing the sampling number leads to a substantial improvement in data quality. Furthermore, leveraging a stronger judge model can effectively compensate for deficiencies in the understanding component, as **shown in Table 3 in A2.**
>
> >**Q7:  Quantitative results on the standard understanding benchmarks of Janus and VILA-U.**
>
> **A7:** Thank you for pointing this out. We provide the experimental results on Janus and VILA-U **in Tables 1 and 2 in A1.**

---

> > ### Comment · Reviewer_zMCR · 2025-08-08
> >
> > Thanks for the response provided by the authors. I will keep my score as positive.

---

> > > ### Author Response · Authors · 2025-08-08
> > >
> > > Dear Reviewer,
> > >
> > > Thank you for keeping positive score. Your support has been invaluable for our research.
> > >
> > > Warm Regards,
> > >
> > > The Authors

---

### Official Review · Reviewer_WWD6 · 2025-07-01

**Clarity:** 4
**Significance:** 2
**Originality:** 3
**Rating:** 4
**Confidence:** 4

**Summary:**

This article studies the performance gap between understanding and generation abilities in MLLMs. The authors observe that unified MLLMs tend to have stronger understanding capabilities than generation, which has not been widely addressed before. To tackle this issue, they introduce HermesFlow, aiming to align these two capabilities. The method uses homologous data (data that shares meaning across modalities) to create preference pairs for both understanding and generation. Then, it applies Pair-DPO and self-play training to improve alignment between the two abilities. Experimental results show that HermesFlow improves both understanding and generation, especially in closing the performance gap.

**Questions:**

As stated in the first or third weakness, why not choosing a SOTA unified MLLM, and using your method to improve its capability to achieve both best and more balanced performance.

**Ethical Concerns:**

["NO or VERY MINOR ethics concerns only"]

**Final Justification:**

I have read the rebuttal from the authors, and raised my score accordingly.

**Limitations:**

yes

**Quality:**

2

**Strengths And Weaknesses:**

**Strengths**:
1. The paper designs a framework to synthesize preference data for both understanding and generation with the target MLLM model and make use of pair-DPO to improve both capabilities iteratively.
2. The paper is clearly written and easy to follow.

**Weaknesses**:
1. Filling the gap between generation and understanding might be a less important problem than improving both of the capabilities. For instance, the improved show-o (the proposed model) still lags behind VILA-U for a large distance in terms of understanding (Table 4), and the performance on generation is maiginally higher, then it maybe still a better choice to use VILA-U.
2. The improvements from self-playiterative method seems limited and marginal.
3. The method might be more useful when applying to less powerful model which is less robust, adding experiment results on larger model like at least 7B to show its generalization and scalability might be a more rogorous setting.

---

> ### Author Rebuttal · Authors · 2025-07-31
>
> *We sincerely thank you for your time and efforts in reviewing our paper and for your valuable feedback. We are glad to see that the paper is well written and easy to follow, and the proposed method is effective. Please see below for our responses to your comments.*
>
> >**Q1: Filling the gap between generation and understanding might be a less important problem than improving both of the capabilities. For instance, the improved show-o (the proposed model) still lags behind VILA-U for a large distance in terms of understanding (Table 4), and the performance on generation is maiginally higher, then it maybe still a better choice to use VILA-U.**
>
> **A1:** I completely agree with your point. However, HermesFlow is the first to empirically demonstrate the **significant gap between understanding and generation**, and it proposes a **general self-improvement post-training method. Without relying on any external supervision data**, it leverages the existing gap to seamlessly align understanding and generation capabilities, while also enabling joint improvement. This makes it a highly generalizable post-training approach that can be applied to **any pretrained unified model** for aligning and enhancing both understanding and generation.
>
> In addition, we argue that a truly unified MLLM should excel in both understanding and generation, **striking a balance rather than favoring one over the other**. However, current unified MLLMs struggle to achieve this equilibrium. To address this, we propose a self-improvement paradigm that effectively bridges the gap, yielding an MLLM performs strongly in both aspects. This balance is crucial in practice: RPG enhances prompt understanding using more powerful MLLM, ultimately improving generation quality; MetaMorph incorporates more generative data, leading to better understanding. This interplay demonstrates that understanding and generation are mutually reinforcing, bridging the gap between them is essential for developing a truly unified MLLM.
>
> Moreover, to validate **the generalization ability of HermesFlow across different architectures and model scales, we reproduced HermesFlow on both Janus (1.3B) and VILA-U (7B)**. The results are shown below:
>
> **Table 1: Evaluation on Multimodal Understanding Benchmarks.**
>
> |                        | #Params | POPE     | MME        | Flickr30k | VQAv2    | GQA      | MMMU     |
> | ---------------------- | ------- | -------- | ---------- | --------- | -------- | -------- | -------- |
> | Show-o                 | 1.3B    | 80.0     | 1232.9     | 67.6      | 74.7     | 61.0     | 27.4     |
> | **HermesFlow(Show-o)** | 1.3B    | **81.4** | **1249.7** | **69.2**  | **75.3** | **61.7** | **28.3** |
> | Janus                  | 1.3B    | 87.0     | 1338.0     | 68.8      | 77.3     | 59.1     | 30.5     |
> | **HermesFlow(Janus)**  | 1.3B    | **87.4** | **1377.3** | **70.2**  | **78.4** | **60.0** | **30.9** |
> | VILA-U                 | 7B      | 85.8     | 1401.8     | 69.9      | 79.4     | 60.8     | 30.1     |
> | **HermesFlow(VILA-U)** | 7B      | **86.9** | **1427.1** | **70.8**  | **79.7** | **61.4** | **31.8** |
>
> **Table 2: Evaluation on Visual Generation Benchmark: GenEval.**
>
> |                        | #Params | Single Obj. | Two Obj. | Counting | Colors   | Position | Color Attri. | Overall  |
> | ---------------------- | ------- | ----------- | -------- | -------- | -------- | -------- | ------------ | -------- |
> | Show-o                 | 1.3B    | 0.98        | 0.77     | 0.58     | 0.81     | 0.23     | 0.44         | 0.64     |
> | **HermesFlow(Show-o)** | 1.3B    | **0.98**    | **0.84** | **0.66** | **0.82** | **0.32** | **0.52**     | **0.69** |
> | Janus                  | 1.3B    | 0.97        | 0.68     | 0.30     | 0.84     | 0.46     | 0.42         | 0.61     |
> | **HermesFlow(Janus)**  | 1.3B    | **0.98**    | **0.89** | **0.48** | **0.87** | **0.56** | **0.55**     | **0.72** |
> | VILA-U                 | 7B      | 0.95        | 0.62     | 0.48     | 0.84     | 0.16     | 0.36         | 0.57     |
> | **HermesFlow(VILA-U)** | 7B      | **0.98**    | **0.88** | **0.65** | **0.85** | **0.39** | **0.53**     | **0.71** |
>
> HermesFlow still shows significant improvements on backbones of larger sizes and different architectures. It can be observed from the table that HermesFlow demonstrates remarkable enhancements in Two Obj., Counting, Position, and Color Attribute on both the 1.3B and 7B backbones. This is because of the excellent understanding ability of MLLM regarding image attributes, spatial position relationships, etc., which makes the preference data contain a large number of features in this regard. This indicates that our architecture has **strong generalization ability and is a general unified model post-training framework**. We will incorporate the relevant experimental updates in the revised manuscript. We appreciate your valuable feedback.
>
>
>
> > **Q2: The improvements from the self-play iterative method seems limited and marginal.**
>
> **A2:** We sincerely apologize for the confusion. Iterative optimization represents a novel paradigm for unified models. HermesFlow shows the most significant performance improvement during the first iteration, which confirms the effectiveness of mutual refinement between understanding and generation, therefore, to continuously narrow the gap between these two aspects, we adopt an iterative training method. Moreover, we observe that the effectiveness of the iterative training method is **highly dependent on the initial gap between understanding and generation** in the backbone model, **the larger the initial gap, the more significant the performance gain from iterative training**. We present the performance metrics of Janus under iterative self-play training:
>
> **Table 3: Effect of Pair-DPO Iterations on Janus**
>
> |                    | #Params | POPE | MME    | MMMU | GenEval(Overall) | DPG-Bench (Average) |
> | ------------------ | ------- | ---- | ------ | ---- | ---------------- | ------------------- |
> | Janus              | 1.3B    | 87.0 | 1338.0 | 30.5 | 0.61             | 73.20               |
> | Pair-DPO (Iter. 1) | 1.3B    | 87.2 | 1365.5 | 30.7 | 0.67             | 75.97               |
> | Pair-DPO (Iter. 2) | 1.3B    | 87.4 | 1363.1 | 30.9 | 0.70             | 76.31               |
> | Pair-DPO (Iter. 3) | 1.3B    | 87.4 | 1377.3 | 30.9 | 0.72             | 77.21               |
>
> As shown in the Table, HermesFlow (Janus) demonstrates more substantial improvements in the second and third iterations. We argue that bridging the gap between understanding and generation is a challenging optimization problem, and a single round of DPO training may not suffice. Therefore, **for models with a larger initial gap, we recommend applying iterative training**. We appreciate your insightful question and will incorporate this important analysis into the revised version of the paper.
>
> > **Q3: The method might be more useful when applying to less powerful model which is less robust, adding experiment results on larger models like at least 7B to show its generalization and scalability might be a more rogorous setting.**
>
> **A3:** Thank you for your suggestion! **In Table 1 and Table 2 in A1**, we present the experimental results of HermesFlow on two models with different architectures and scales: Janus (1.3B) and VILA-U (7B). We will incorporate the relevant experimental updates in the revised manuscript. We appreciate your valuable feedback on this matter.
>
> > **Q4: As stated in the first or third weakness, why not choose a SOTA unified MLLM, and use your method to improve its capability to achieve both best and more balanced performance.**
>
>  **A4:** HermesFlow is an initial attempt, and we have already validated its effectiveness on Show-o, Janus, and VILA-U  **In Table 1 and Table 2 in A1**.  We are committed to continuing experiments of HermesFlow on more backbone models (e.g., Bagel and Show-o2) to achieve stronger unified performance.

---

> > ### Author Response · Authors · 2025-08-04
> > **Gentle Reminder**
> >
> > Dear reviewer:
> >
> > We sincerely appreciate the time and effort you dedicated to reviewing our paper. In response to your concerns, we have conducted additional experiments and provided an in-depth analysis on our method.
> >
> > As the discussion period concludes in two days, we kindly request, if possible, that you review our rebuttal at your convenience. Should there be any further points requiring clarification or improvement, please know that we are fully committed to addressing them promptly. Thank you once again for your invaluable contribution to our research.
> >
> > Warm regards,
> >
> > The Authors

---

> > > ### Comment · Reviewer_WWD6 · 2025-08-04
> > >
> > > Thanks for your rebuttal, which mostly addressed my concerns.

---

> > > > ### Author Response · Authors · 2025-08-04
> > > >
> > > > Dear Reviewer WWD6:
> > > >
> > > > We are very pleased that our rebuttal has addressed your concerns. Your feedback has been invaluable for our research.
> > > >
> > > > Warm regards,
> > > >
> > > > The Authors

---

### Official Review · Reviewer_A63A · 2025-07-03

**Clarity:** 2
**Significance:** 2
**Originality:** 3
**Rating:** 4
**Confidence:** 4

**Summary:**

This paper studies an interesting topic: bridging the gap between visual understanding and generation of a unified MLLM through reinforcement learning. One observation is that current unified models usually perform better on image understanding than image generation tasks. Therefore, this paper proposes to use Pair-DPO and self-play iterative opetimization to close the gap between the two tasks. The evaluation on understanding and generation tasks verify its effectiveness.

**Questions:**

see weaknesses.

**Ethical Concerns:**

["NO or VERY MINOR ethics concerns only"]

**Final Justification:**

I increase my score to 4. But I still find it very strange that the authors did not test the results of their own baseline method. This has also shifted my rating from initially positive to negative.

**Limitations:**

yes

**Quality:**

3

**Strengths And Weaknesses:**

**Strengths:**
- The motivation is clear. Current unified models do good on understanding but are limited to their visual generation ability. Therefore, a method to mitigating the gap between two tasks is important for future research.
- The proposed method is interesting. HermesFlow constructs preference pairs by comparing the generated text/image with training image-caption pair. Then Pair-DPO is performed on both image understanding and generation tasks. The policy MLLM is optimized to generate text/image that aligned better with the training image-caption pairs.

**Weaknesses:**
- Limited improvement on benchmarks, such as Tab.1 multimodal understanding, Tab.2 image generation.
- As for the comparisons with show-o on Geneval. I am confused which version of show-o is used, 256px or 512px? In Tab.2, the baseline show-o is 0.53 on Geneval, but I guess HermesFlow use 512px show-o from the source code. So the comparison is not fair in Tab.2 and Tab.5. The real gain is 0.68 -> 0.69, not 0.53 -> 0.69.

---

> ### Author Rebuttal · Authors · 2025-07-31
>
> *We sincerely thank you for your time and efforts in reviewing our paper and for your valuable feedback. We are glad to see that the motivation of our paper is clear, the topic is interesting and important, the proposed method is effective. Please see below for our responses to your comments.*
>
> > **Q1: Limited improvement on benchmarks, such as Tab.1 multimodal understanding, Tab.2 image generation.**
>
> **A1:** Sorry for the confusion. This paper investigates whether **self-improvement and self-alignment without any external supervision data** can be realized in unified multimodal LLMs, particularly addressing the existing gap between understanding and generation capabilities. While our current work presents an initial approach, we aim to explore more refined optimization methods in future research to further advance performance.
>
> Moreover, to validate **the generalization ability of HermesFlow across different architectures and model scales, we reproduced HermesFlow on both Janus (1.3B) and VILA-U (7B)**. The results are shown below:
>
> **Table 1: Evaluation on Multimodal Understanding Benchmarks.**
>
> |                        | #Params | POPE     | MME        | Flickr30k | VQAv2    | GQA      | MMMU     |
> | ---------------------- | ------- | -------- | ---------- | --------- | -------- | -------- | -------- |
> | Show-o                 | 1.3B    | 80.0     | 1232.9     | 67.6      | 74.7     | 61.0     | 27.4     |
> | **HermesFlow(Show-o)** | 1.3B    | **81.4** | **1249.7** | **69.2**  | **75.3** | **61.7** | **28.3** |
> | Janus                  | 1.3B    | 87.0     | 1338.0     | 68.8      | 77.3     | 59.1     | 30.5     |
> | **HermesFlow(Janus)**  | 1.3B    | **87.4** | **1377.3** | **70.2**  | **78.4** | **60.0** | **30.9** |
> | VILA-U                 | 7B      | 85.8     | 1401.8     | 69.9      | 79.4     | 60.8     | 30.1     |
> | **HermesFlow(VILA-U)** | 7B      | **86.9** | **1427.1** | **70.8**  | **79.7** | **61.4** | **31.8** |
>
> **Table 2: Evaluation on Visual Generation Benchmark: GenEval.**
>
> |                        | #Params | Single Obj. | Two Obj. | Counting | Colors   | Position | Color Attri. | Overall  |
> | ---------------------- | ------- | ----------- | -------- | -------- | -------- | -------- | ------------ | -------- |
> | Show-o                 | 1.3B    | 0.98        | 0.77     | 0.58     | 0.81     | 0.23     | 0.44         | 0.64     |
> | **HermesFlow(Show-o)** | 1.3B    | **0.98**    | **0.84** | **0.66** | **0.82** | **0.32** | **0.52**     | **0.69** |
> | Janus                  | 1.3B    | 0.97        | 0.68     | 0.30     | 0.84     | 0.46     | 0.42         | 0.61     |
> | **HermesFlow(Janus)**  | 1.3B    | **0.98**    | **0.89** | **0.48** | **0.87** | **0.56** | **0.55**     | **0.72** |
> | VILA-U                 | 7B      | 0.95        | 0.62     | 0.48     | 0.84     | 0.16     | 0.36         | 0.57     |
> | **HermesFlow(VILA-U)** | 7B      | **0.98**    | **0.88** | **0.65** | **0.85** | **0.39** | **0.53**     | **0.71** |
>
> HermesFlow still shows significant improvements on backbones of larger sizes and different architectures. In addition, it can be observed that HermesFlow demonstrates remarkable enhancements in Two Obj., Counting, Position, and Color Attribute on both the 1.3B and 7B backbones. This is because of the excellent understanding ability of MLLM regarding image attributes, spatial position relationships, etc., which makes the preference data contain a large number of features in this regard.
>
> Moving beyond the framework limitations of model self-improvement without any external supervision, we use **Gemini-2.5-Pro as a judge** to collect preference datasets from multiple generated data samples:
>
> **Table 3: Use Gemini-2.5-Pro as the Judge to Create Preference Data**
>
> |                               | #Params | POPE | MME    | MMMU | GenEval(Overall) | DPG-Bench (Average) |
> | ----------------------------- | ------- | ---- | ------ | ---- | ---------------- | ------------------- |
> | Show-o                        | 1.3B    | 80.0 | 1232.9 | 27.4 | 0.64             | 67.48               |
> | HermesFlow (Self-improvemrnt) | 1.3B    | 81.4 | 1249.7 | 28.3 | 0.69             | 70.22               |
> | HermesFlow (MLLM-as-a-judge)  | 1.3B    | 81.7 | 1258.4 | 28.7 | 0.73             | 70.99               |
>
> From the table, employing a more powerful external model as a data collector yields better results in both understanding and generation. This is because Gemini-2.5-Pro is capable of thoroughly mining and selecting the optimal win and lose samples, which also highlights the potential of the HermesFlow architecture.
>
> We will incorporate the relevant experimental updates in the revised manuscript. We appreciate your valuable feedback on this matter.
>
>
>
>
>
> > **Q2: As for the comparisons with show-o on Geneval. I am confused which version of show-o is used, 256px or 512px? In Tab.2, the baseline show-o is 0.53 on Geneval, but I guess HermesFlow use 512px show-o from the source code. So the comparison is not fair in Tab.2 and Tab.5. The real gain is 0.68 -> 0.69, not 0.53 -> 0.69.**
>
>
>
> **A2:** We sincerely appreciate your careful observation and bringing this important point to our attention. We referred to papers Janus and Emu3, and followed their reported value of 0.53 without carefully verifying the resolution setting. HermesFlow used the 512px version. We sincerely apologize for this oversight, as we did not rigorously check this value.
>
> Here, we re-evaluate the performance on Show-o using the following configuration: **guidance scale = 5**, **generation timesteps = 50**, and **resolution = 512px**. The updated results are as follows:
>
> **Table 4: Comparison between Show-o and HermesFlow on GenEval.**
> |            | #Params | Single Obj. | Two Obj.    | Counting    | Colors      | Position    | Color Attri. | Overall     |
> | ---------- | ------- | ----------- | ----------- | ----------- | ----------- | ----------- | ------------ | ----------- |
> | Show-o     | 1.3B    | 0.98        | 0.77        | 0.58        | 0.81        | 0.23        | 0.44         | 0.64        |
> | HermesFlow | 1.3B    | 0.98(+0.00)  | 0.84(+0.07) | 0.66(+0.08) | 0.82(+0.01) | 0.32(+0.09) | 0.52(+0.08)  | 0.69(+0.05) |
>
> HermesFlow achieved significant improvements, particularly on Two Obj., Counting, Position, and Color Attribute. This is because the **understanding of MLLMs exhibits a clear advantage, especially in handling multi-object and spatial relationship scenarios, where it can easily identify and select high-quality preference data**. This further validates the existence of a generation-understanding gap in MLLMs, indicating clear room for improvement.
>
> We sincerely appreciate your careful observation of this issue. We will revise our paper accordingly and thoroughly verify the correctness of other reported metrics.

---

> > ### Author Response · Authors · 2025-08-04
> > **Gentle Reminder**
> >
> > Dear reviewer:
> >
> > We sincerely appreciate the time and effort you dedicated to reviewing our paper. In response to your concerns, we have conducted additional experiments and provided an in-depth analysis on our method.
> >
> > As the discussion period concludes in two days, we kindly request, if possible, that you review our rebuttal at your convenience. Should there be any further points requiring clarification or improvement, please know that we are fully committed to addressing them promptly. Thank you once again for your invaluable contribution to our research.
> >
> > Warm regards,
> >
> > The Authors

---

> > ### Comment · Reviewer_A63A · 2025-08-07
> >
> > I have read the response and believe it resolved my concerns. I will increase my score to 4. But I still find it very strange that the authors did not test the results of their own baseline method. This has also shifted my rating from initially positive to negative.

---

> > > ### Author Response · Authors · 2025-08-08
> > >
> > > Dear Reviewer,
> > >
> > > Thank you very much for increasing your score to 4 and for acknowledging that our response resolved your concerns.
> > > We completely understand your surprise regarding the baseline verification issue. We sincerely apologize for this oversight. Our initial intention was to maintain consistency with previously reported values from existing papers (Janus and Emu3) to ensure fair comparison standards. However, we recognize that we should have independently verified these baseline results regardless of prior reports.
> > >
> > > Importantly, even after correcting the baseline to the proper 512px Show-o results, HermesFlow still demonstrates consistent and meaningful improvements across multiple challenging aspects, particularly in multi-object handling, counting, and spatial relationships. This validates the effectiveness of our approach and confirms that the improvements are genuine rather than artifacts of mismatched experimental settings.
> > >
> > > We will update these results in our revised manuscript and ensure rigorous verification. Thank you again for your careful review - it has significantly improved the quality and reliability of our work.
> > >
> > > Best regards,
> > >
> > > The Authors

---

> ### Author Response · Authors · 2025-08-07
> **Third Follow-up to Reviewer A63A**
>
> Dear Reviewer A63A,
>
> We hope our comprehensive responses and additional experiments have addressed your concerns. Given the approaching deadline, could you please provide your updated evaluation or let us know if you need any further clarification?
>
> Thank you for your time and consideration.
>
> Best regards,
>
> The Authors

---

### Official Review · Reviewer_KKjB · 2025-07-04

**Clarity:** 2
**Significance:** 3
**Originality:** 3
**Rating:** 4
**Confidence:** 2

**Summary:**

This paper identified a performance gap between multimodal understanding and generation, and proposed a framework to reduce this gap.

**Questions:**

What are the designing philosophies of the generation score and understanding score? Are there any alternative approaches to implement them?

**Ethical Concerns:**

["NO or VERY MINOR ethics concerns only"]

**Final Justification:**

Although the VQA-based evaluation is my biggest concern, but this paper is introducing a new important problem and conducted comprehensive experiments. I vote for a borderline acceptance.

**Limitations:**

Discussed briefly in conclusion.

**Paper Formatting Concerns:**

No concern

**Quality:**

2

**Strengths And Weaknesses:**

Strengths:
- Strong motivation. The problem of exploring the performance gap between multimodal understanding and generation is important and under-explored.
- The proposed method successfully reduced the gap between understanding score and generation score.

Weaknesses:
- The definition of generation score may bring noises and may be hacked by 4o model. The generation score, which is important for calculation the performance gap, is heavily relied on the gpt 4o model to answer questions based on generated images. However, it's unclear how these visual information are important in answering these questions. Thus this score may not faithfully reflect the generation performance. Some more widely-used generation power metric or more grounded metrics may be needed.
- In Table 5, GenEval of Pair-DPO (Iter. 3)  should not be bolded as it is not the best one.
- Figure 3 is difficult to understand as it lacks some visual orders or focuses. Some interaction or mentioning of Fig 3 in the texts may be needed. It can also be removed if the main information are all included in the text.
- Results are largely missing in Table 1.
- The paper lacks evaluation on the quality of the grouped winning and losing samples, and how they differ.

---

> ### Author Rebuttal · Authors · 2025-07-30
>
> *We sincerely thank you for your time and efforts in reviewing our paper and for your valuable feedback. We are glad to see that the motivation of our paper is strong and important, and the proposed method is effective. Please see below for our responses to your comments.*
>
>
> > **Q1: The definition of generation score may bring noises. It's unclear how these visual information are important in answering these questions.**
>
> **A1:** Sorry for the confusion. The generation score refers to the average score computed by GPT-4o through Visual Question Answering. The VQA questions used here are limited to **basic visual perception tasks**, such as object existence, attributes, and correctness of spatial relationships. GPT-4o has a **significant advantage in handling these relatively simple perception-based questions**. Therefore, using GPT-4o to compute the generation score does not introduce significant errors. In addition, we also conducted experiments **using Gemini-2.5-Pro(0605)**, the current state-of-the-art vision-language model. The results are as follows:
>
> **Table1: Comparison between Using GPT-4o and Gemini2.5-Pro to Calculate Generation Score.**
> |Method|# Params|Generation Score(GPT-4o)|Generation Score(Gemini-2.5-Pro)|
> |-|-|-|-|
> |VILA-U|7B|0.477|0.473|
> |Janus|1.3B|0.417|0.410|
> |Show-o|1.3B|0.433|0.438|
> |HermesFlow|1.3B|0.497| 0.495|
>
> Since these VQA questions focus on simple visual perception tasks, the experimental results show that GPT-4o and Gemini-2.5-Pro exhibit very similar performance. Therefore, using GPT-4o to calculate the generation score does not introduce significant errors. Thank you for your valuable feedback.
>
> > **Q2: In Table 5, GenEval of Pair-DPO (Iter. 3) should not be bolded as it is not the best one.**
>
> **A2:** Thank you for pointing out this issue. We will update the manuscript and ensure that similar problems do not appear in other parts of the paper.
>
> > **Q3: Figure 3 is difficult to understand as it lacks some visual orders or focuses.**
>
>  **A3:** We apologize for any inconvenience caused. We will make sure to present Figure 3 more clearly in the revised manuscript, including aspects such as visual order and focus. Here, I’d like to help clarify the workflow of Figure 3:
>
> HermesFlow consists of two main components:
>
> - The curation of homologous preference data (left half of Figure 3)
> - Pair-DPO training (right half of Figure 3)
>
> Regarding the construction of homologous preference data, we **use homologous data (i.e., image-caption pairs) as input**. Given the image, we use the MLLM's understanding capability to perform the image captioning task, forming the **understanding pair**. Given the image caption, we use the MLLM's generation capability to perform the text-to-image task, forming the **generation pair**. In this way, the MLLM’s understanding and generation branches collaboratively construct the homologous preference data from the input pair.
>
> Based on this preference data, we propose a **Pair-DPO training strategy** to optimize the backbone model. Meanwhile, we design an **iterative self-play** mechanism (illustrated by the red dashed arrows in the figure) to continuously update the preference dataset in a dynamic fashion—gradually aligning and enhancing both the model’s understanding and generation capabilities.
>
> Thank you again for your valuable feedback. We will redraw Figure 3 to make it clearer and easier to understand. If you still have any questions, please feel free to let us know.
>
> > **Q4: Results are largely missing in Table 1.**
>
>  **A4:** This was an oversight on our part. Table 1 references results from the VILA-U, Janus, and Show-o. We have now completed most experiments for Table 1 and removed redundant baseline models. Moreover, to validate **the generalization ability of HermesFlow across different architectures and model scales, we reproduced HermesFlow on both Janus (1.3B) and VILA-U (7B)**. The results are shown below:
>
> **Table 2: Evaluation on Multimodal Understanding Benchmarks.**
>
> | | #Params|POPE| MME| Flickr30k | VQAv2| GQA| MMMU|
> |-|-|-|-|-|-|-|-|
> | Emu|13B|80.9|1289.2|77.4| 57.2| 57.7| 25.7|
> | NExT-GPT|13B|82.3 |1222.1| 84.5| 66.7 | 60.2| 25.5|
> | SEED-X |13B|84.2|1435.7| 52.3| -| 47.9| 35.6 |
> | Chameleon| 34B|78.9|-| 74.7| 66.0| 54.4 | 29.5|
> | Show-o | 1.3B| 80.0|1232.9| 67.6| 74.7| 61.0 | 27.4|
> | **HermesFlow(Show-o)** |1.3B|**81.4**|**1249.7**| **69.2** |**75.3** | **61.7** | **28.3** |
> | Janus| 1.3B| 87.0| 1338.0 | 68.8| 77.3 | 59.1| 30.5|
> | **HermesFlow(Janus)**  | 1.3B | **87.4** | **1377.3**|**70.2** |**78.4**|**60.0** | **30.9** |
> | VILA-U | 7B | 85.8| 1401.8 | 69.9| 79.4| 60.8 | 30.1|
> | **HermesFlow(VILA-U)** | 7B| **86.9** |**1427.1**|**70.8** | **79.7**|**61.4** | **31.8** |
>
> **Table 3: Evaluation on Visual Generation Benchmark: GenEval.**
>
> | | #Params | Single Obj. | Two Obj. | Counting | Colors | Position | Color Attri. | Overall  |
> | -| -| - | -| - | - | - | - | -|
> | Show-o | 1.3B| 0.98|0.77|0.58|0.81|0.23|0.44|0.64|
> | **HermesFlow(Show-o)** | 1.3B | **0.98**| **0.84**|**0.66**|**0.82**|**0.32**|**0.52**|**0.69**|
> | Janus|1.3B|0.97|0.68|0.30|0.84|0.46|0.42|0.61|
> | **HermesFlow(Janus)** | 1.3B | **0.98**| **0.89** | **0.48** | **0.87** | **0.56** | **0.55** | **0.72**|
> | VILA-U  | 7B | 0.95| 0.62 | 0.48| 0.84 | 0.16 | 0.36| 0.57|
> | **HermesFlow(VILA-U)** | 7B  | **0.98**  | **0.88** | **0.65** | **0.85** | **0.39** | **0.53** |**0.71**|
>
> HermesFlow still shows significant improvements on backbones of larger sizes and different architectures. This indicates that our architecture has **strong generalization ability and is a general unified model post-training framework**. We will incorporate the relevant experimental updates in the revised manuscript. We appreciate your valuable feedback.
>
> > **Q5: The paper lacks evaluation on the quality of the grouped winning and losing samples, and how they differ.**
>
> **A5:** We thank the reviewer for raising this important point. We provide a comparison between the win and lose samples in terms of both understanding and generation aspects:
>
> **Table 4: Comparison between Win Samples and Lose Samples.**
>
> |   | Count | Mean Caption Length | Und: BERT Similarity | Und: Model Judge | Gen: CLIP-Score | Gen: VQA-Score |
> | - | - | - | -| -| -| -|
> | Win samples  | 5K| ~26.3 words | 0.673| 79.2% | 31.11| 0.482 |
> | Lose samples | 5K| ~18.8 words| 0.488| 20.8%  | 29.98 | 0.392 |
>
> **Regarding the understanding data of image captions**, we use BERT Similarity and the win rate of the Model Judge (using Gemini-2.5-Pro) to evaluate the caption quality of win samples and lose samples. It is clear that captions of win samples have more accurate image descriptions. Moreover, our analysis of mean caption length reveals that captions in win samples tend to be longer, and therefore provide more detailed descriptions.
>
> **Regarding the generation data of generated images**, we use CLIP-Score and VQA-Score (using Gemini-2.5-Pro) for evaluation. This indicates that win samples exhibit higher quality in image-text alignment. It also suggests that leveraging the unified model’s understanding capabilities during generation enables self-filtering of higher-quality outputs. We appreciate your suggestion and will update the paper to include this important analysis.
>
> > **Q6: What are the designing philosophies of the generation score and understanding score? Are there any alternative approaches to implement them?**
>
> **A6:** Since understanding and generation are two entirely distinct tasks, existing benchmarks evaluate these two aspects separately, employing different evaluation criteria and assessment methods. Consequently, **it is difficult to quantitatively verify the relative strengths between understanding and generation.** To address this gap where no unified standard exists to measure both understanding and generation, we propose the method shown in Figure 2(a) of our paper and provide a detailed explanation in the paragraph at line 239.
>
> It is a highly promising and meaningful direction to explore more robust methods to uniformly measure understanding and generation. Here we propose a novel evaluation approach:
>
> We consider the conflict between understanding and generation. For an input prompt $p$, we define the generated image as $\phi^{Gen}(p)$. Using the understanding part of MLLM to judge whether this generated image satisfies the prompt, we define the **contradiction score** as:
> $$
> S_c = \frac{1}{|\mathcal{P}|}\sum_{p\in\mathcal{P}}\mathbb{I}[\phi^{Und}(\phi^{Gen}(p), query)=0]
> $$
> $$
> Judge_\phi=\phi^{Und}(\phi^{Gen}(p), query)
> $$
> Where $\mathcal{P}$ represents the prompt dataset, and $query$ denotes the judge prompt for understanding. Using Gemini-2.5-Pro as the supervised model, the understanding score and generation score can be respectively defined as follows:
> $$
> S_{Gen} = 1-\frac{1}{|\mathcal{P}|}\sum_{p\in\mathcal{P}}\mathbb{I}( Judge_\phi = 0 \land Judge_{Gemini} = 0 )
> $$
> $$
> S_{Und} = 1-\frac{1}{|\mathcal{P}|}\sum_{p\in\mathcal{P}}\mathbb{I}( Judge_\phi = 0 \land Judge_{Gemini} = 1 )
> $$
> Using this approach, we conducted new experiments and obtained the following results:
>
> **Table5: Comparison between Method1 (used in the paper) and Method2 (the newly proposed).**
>
> | Method     | # Params | Generation Score(method1) | Understanding Score(method1) | Generation Score(method2) | Understanding Score(method2) |
> | - | -| - | -| -| -|
> | VILA-U | 7B | 0.473| 0.646| 0.622| 0.901|
> | Janus | 1.3B | 0.410| 0.599| 0.607| 0.927 |
> | Show-o | 1.3B | 0.438| 0.520| 0.644| 0.893  |
> | HermesFlow | 1.3B| 0.495| 0.533| 0.770| 0.912 |
>
> This table demonstrates that both the new method and the original method in the paper can effectively measure the gap between understanding and generation in the unified model. Notably, the new method provides more pronounced and amplified measurements of this gap. We sincerely appreciate your insightful suggestion and will incorporate it into the updated version of our paper.

---

> > ### Comment · Reviewer_KKjB · 2025-08-05
> >
> > Thanks for the rebuttal. To better understand the evaluation, I am following up on Q1: Is such VQA-based evaluation created by you or established by other work?

---

> > > ### Author Response · Authors · 2025-08-06
> > > **Official Comment by Authors**
> > >
> > > Dear Reviewer KKjB:
> > >
> > > Thank you for raising such a valuable question. The VQA-based method is our own proposal. Our initial idea was inspired by the general insight from UniRL[1], which suggests that **understanding and generation can be evaluated in a sequential manner**. For the same image or prompt, MLLMs often struggle to achieve consistent understanding and generation. Building on this observation, we propose a unified evaluation scheme based on a contradiction score, using Gemini-2.5-Pro as a strong evaluator. This design aims to effectively capture and evaluate the inconsistency in MLLM behavior.
> > >
> > > We will update our paper accordingly, including a more detailed discussion of this approach and its relation to prior work in the *Related Work* section.
> > >
> > >
> > >
> > > [1] UniRL: Self-Improving Unified Multimodal Models via Supervised and Reinforcement Learning
> > >
> > > Warm regards,
> > >
> > > The Authors

---

> > > > ### Comment · Reviewer_KKjB · 2025-08-07
> > > >
> > > > Thanks for your clarifications. I would like to point out that some more fundamental metrics may be needed because of VQA is just one possible downstream task forms of multimodal tasks. But I also understand that there is no established evaluation methods. I keep my positive score.

---

> > > > > ### Author Response · Authors · 2025-08-08
> > > > >
> > > > > Dear Reviewer,
> > > > >
> > > > > Thank you for maintaining your positive evaluation and for your constructive feedback regarding evaluation metrics.
> > > > > Your point about incorporating more fundamental metrics beyond VQA is well-taken, and we will consider expanding our evaluation framework in future work to provide a more comprehensive assessment across diverse multimodal task forms.
> > > > >
> > > > > We are grateful for your valuable insights throughout the review process.
> > > > >
> > > > > Best regards,
> > > > >
> > > > > The Authors

---

### Decision · Program_Chairs · 2025-09-17

**Decision:**

Accept (poster)

**Comment:**

The reviewers agree that the proposed HermesFlow addresses a critical issue of bridging the gap between understanding and generation capabilities for a unified multimodal LLM. The proposed Pair-DPO and self-play iterative optimization techniques are novel and effective, as evidenced by the experimental results. The authors are also actively engaged in the rebuttal phase and have addressed the majority of concerns raised by the reviewers. We recommend that the authors incorporate the discussions into the next version of the work and expand the evaluation to encompass a broader range of benchmarks.